# The Curious Case of the Choledochal Cyst—Revisiting the Todani Classification: Case Report and Review of the Literature

**DOI:** 10.3390/diagnostics13061059

**Published:** 2023-03-10

**Authors:** Adrian Miron, Liliana Gabriela Popa, Elena Adelina Toma, Valentin Calu, Radu Florin Parvuletu, Octavian Enciu

**Affiliations:** 1Faculty of Medicine, Carol Davila University of Medicine and Pharmacy, 050474 Bucharest, Romaniaoctavian.enciu@umfcd.ro (O.E.); 2Elias Emergency University Hospital, 011461 Bucharest, Romania

**Keywords:** choledochal cysts, Todani classification, hepaticojejunostomy

## Abstract

Choledochal cysts (CCs) are rare occurrences presenting as dilatations of biliary structures, which can present as single or multiple dilatations and can appear as both intra- and extrahepatic anomalies. The most widespread classification of CCs is the Todani classification, but there have been numerous reports of cysts that do not fall into any of the types described. We present such a case—a male patient 36 years of age who underwent preoperative CT, MRCP, and ERCP, which mistakenly indicated a type II Todani CC, and intraoperatively was found to be located at the confluence of the hepatic ducts and encompassed the origin of the common bile duct. Complete resection of the cyst and the proximal segment of the common bile duct was performed, and reconstruction was carried out by Roux-en-Y double-tutorized hepaticojejunostomy. Considering the risk of malignant transformation, the frequent preoperative misdiagnosis, as well as the technically challenging surgery required in such cases, we advocate for a revision of the classification and raise awareness of the need for guidelines regarding the proper short-term and long-term management of this disease to ensure adequate quality of life and disease-free survival for patients.

## 1. Introduction

Choledochal cysts (CCs) are uncommon congenital dilatations of the biliary tree, affecting its extrahepatic or intrahepatic segments or both at single and multiple sites. Most of them interest the common bile duct. CCs represent the second most frequent inborn biliary duct anomaly after biliary atresia [1].

The incidence of CCs varies greatly around the globe, being estimated at 1:100,000–1:150,000 in Western countries, 1:13,500 in the US, 1:15,000 in Australia, and up to 1:1000 in Asian populations [2,3,4,5]. Moreover, approximately 2/3 of the cases arising in Asia have been reported in Japan [3]. CCs also have a predilection for the female gender, the anomaly being 3–4 times more frequent in women [6]. The geographical and gender differences in CC incidence remain unexplained.

Patients are usually diagnosed with CCs during childhood, but in 20–25% of cases, the diagnosis is delayed until adulthood [6,7]. This poses real threats as CCs are precancerous lesions with a malignancy rate that ranges from 2.5% to 28%, according to different reports, and increases with age [8,9]. Moreover, postoperative complications can be serious and difficult to manage, and delayed biliary complications can arise as late as two decades after surgery [10]. 

The first mention of CCs dates back to 1723 when Vater and Ezler discussed abnormal dilations of the biliary ducts. CCs were first thoroughly described in 1959 by Alonso-LEJ et al., who attributed the term choledochal cysts and classified them into three types based on the anatomic location and morphology of the cyst. Todani et al. added two more types and several subtypes to this classification in 1977, which they updated in 1997 and 2003 [11]. 

The most frequent type of CC is type I (50%–80% of all cases), followed by type IV (12–35%), type V (20%), type III (1.4–4.5%), and type II (2–3%) [5,12]. 

Type I CCs encompass three subtypes, depending on the site and morphology of the cyst. While in type IA, the cystic dilation involves the entire extrahepatic biliary system, it only affects a segment of the extrahepatic biliary tree in type IB. Type IC is characterized by a fusiform dilatation of the entire extrahepatic biliary system, as well as the dilation of the intrahepatic ducts. Type ID is a proposed subtype that we encountered in the case described below. It is a cystic extrahepatic fusiform dilatation at the biliary confluence, with a non-dilated intrahepatic biliary tree, non-dilated distal CBD, and no pancreaticobiliary maljunction.

Type II CCs represent a saccular diverticulum of the common bile duct. 

Type III CCs or choledochocele consist of an intramural dilation of the distal segment of the common bile duct, outpouching into the duodenal wall. It is often accompanied by ampullary obstruction.

Type IV CCs include two subtypes. Type IVA displays intra- and extrahepatic dilation of the biliary ducts, whereas in the less common type IVB, only the extrahepatic biliary tree is dilated at multiple sites. 

Type V CCs, also termed Caroli’s disease, are defined by the presence of single or multiple dilatations confined to the intrahepatic bile ducts. If the patient also suffers from congenital hepatic fibrosis, the association is referred to as Caroli’s syndrome [5,12] (Figure 1).

Types IA, IC, and IVA are usually associated with a pancreaticobiliary maljunction [12]. Types I and IV have been repeatedly cited in both case reports and systematic reviews as having the highest potential for long-term postoperative complications, including intrahepatic bile duct dilatation, the development of intrahepatic calculi, cholangitis, strictures, and metachronous malignancy [10,13,14].

Although Todani’s classification is still generally used in clinical practice, many experts argue it should be revised and refined, considering the numerous reports of unusual types of CCs and new insights into their pathogenesis.

We present the case of a male adult patient diagnosed with a CC whose location and morphology do not fall into any of the categories listed above and review the literature regarding the variants of CCs that have not been included in the currently used classification.

## 2. Case Report

A 35-year-old male patient was referred to our clinic for a 5-day history of jaundice, fever, and abdominal pain. The physical examination revealed scleral icterus and right upper quadrant tenderness without inspiratory arrest at palpation (absent Murphy’s sign).

On admission, laboratory workup showed elevated serum levels of total bilirubin (6.2 g/dL), direct bilirubin (5.3 g/dL), lipase (1029 U/L, more than threefold the normal upper limit), and impaired liver function (ALT 328 U/L, AST 221 U/L). On the third day of hospitalization, the serum level of total bilirubin reached 14.6 g/dL, the value of direct bilirubin was 13.4 g/dL, and that of lipase 33 U/L, while the liver function tests remained elevated (AST 205 U/L, ALT 259 U/L).

An initial abdominal ultrasound detected gallstone disease, but the gallbladder had thin walls and an oval-shaped proximal biliary duct dilatation measuring a maximum of 12 mm diameter. The computed tomography (CT) scan demonstrated nondilated intrahepatic biliary ducts and a proximal common bile duct (CBD) cystic dilatation of 16 mm in the anterior-posterior diameter but without distal (retroduodenal) dilatation, the CBD measuring 5.5 mm in this section, with no focal liver lesions; the CBP did not contain calculi (Figure 2A). The CT description matched a type II CC. For a more detailed anatomy of the biliary tree and the type of CC, a magnetic resonance cholangiopancreatography (MRCP) was requested—a 2.1/1.9/2.3 cm cystic lesion was described at the level of the hepatic hilum, separated only by an extremely thin septum from the CBD. MRCP did not describe pancreaticobiliary maljunction (Figure 2B).

The imaging studies were completed with an endoscopic retrograde cholangiopancreatography (ERCP). The imaging studies we had performed after admission suggested a type II CC and since the patient was jaundiced, and we had planned a laparoscopic resection following ERCP and the endoscopic implantation of a stent, we deemed the benefit of performing an invasive procedure outweighed the risks. The desired outcome was to stent the CBP in order to facilitate the laparoscopic resection. However, contrary to the previous imaging investigations, the ERCP revealed a proximal fusiform CBD dilation that received the right and left hepatic ducts; both the right and left hepatic ducts and the distal CBP were not dilated; their cannulation with the guide wire was unsuccessful. No specific complication was encountered after the endoscopic procedure (Figure 2C).

With the presumptive diagnosis of type II CC in mind, laparoscopic diverticulectomy was planned. During laparoscopy, the cystic dilation was found to be located at the confluence of the hepatic ducts and origin of the CBD, a highly unusual occurrence not included in Todani’s classification; the cyst had an evident extrahepatic localization, with both hepatic ducts and distal CBD duct having normal calibers and a long cystic duct joined the CBP well below the dilatation, in an apparently normal section (Figure 3A–C). Conversion to open surgery was deemed necessary to achieve complete resection of the cyst and the proximal segment of the common bile duct. Reconstruction was carried out by Roux-en-Y double-tutorized hepaticojejunostomy (Figure 3D). The patient’s postoperative course was uneventful. Blood tests and repeated abdominal ultrasound scans were all normal by the 5th postoperative day, and the patient was discharged 2 days later.

The resection specimen is demonstrated in Figure 4A,B—fusiform CBD dilatation with thick but even walls that received both the right and left hepatic ducts—as it appears, a cystic dilatation of the biliary confluence. The pathology report described chronic gallbladder inflammation; the cystic wall was formed by biliary columnar epithelium with focal ulcerations and marked transmural polymorphic inflammatory infiltration. (Figure 4C). The CBP lesion was considered compatible with congenital dilatation of the common bile duct.

Sequential 6-month follow-ups for 15 months postoperatively showed no complications, no change in laboratory test results, including CA 19-9, no stricture or fistula of the anastomoses, and no other dilatations of the hepatic ducts. The patient is scheduled for a long-term follow-up in our clinic. 

## 3. Discussion

The medical literature is abundant in case reports of atypical, uncategorized bile duct cysts, and many experts recommend the revision of CCs classification. 

A “forme fruste” of CC was described by Lilly et al. in 1985 and later reported by other authors as well, all of whom support its inclusion in the classification of CCs [15]. It is characterized by an abnormal pancreaticobiliary ductal junction (APBDJ) in the absence of bile tract cysts, with identical clinical manifestations with the latter and a high risk of gallbladder cancer [15,16]. 

In 2008, Calvo-Ponce et al. reported the case of a patient with an isolated cyst of the common hepatic duct and proposed the addition of this CC variant to Todani’s classification as type ID [17]. To our knowledge, our case is the second of this type published so far. This type of cyst is challenging for the surgeon and requires expertise and adequate preoperative planning, which is why a revised classification of the types of CCs should be considered. 

Other atypical findings are represented by diverticular cysts arising from CCs. Four such cases (1.1% in their study that included 356 patients) of type II diverticulum originating in a CC type IC were reported by Kaneyama et al., who considered them mixed type I and II CCs [18], and another four were reported by Loke et al. [19].

Upon studying 39 adult patients with CCs, Visser et al. concluded that all type I CCs displayed some degree of intrahepatic dilation and questioned the utility of Todani’s classification, deeming the differentiation between CCs type I and IV arbitrary [20].

Choledochoceles and Caroli’s disease are considered by many authors distinct entities that, despite their resemblance to CCs, are not related to the latter and should not be included in the classification of CCs [5,21].

Cystic duct cysts are rare forms of CCs that have also been omitted from the traditional CCs classification. Their clinical and histopathological features are consistent with CCs. Since the first report of a cystic duct cyst published by Stoppa et al. in 1965, many followed, and Serradel et al. designated them as a ‘‘type VI choledochal cyst” [22,23]. Cases of both solitary fusiform or saccular cystic duct dilations and cystic duct cysts associated with CCs in other locations have thereafter been described [24,25].

Michaelides et al. reported six cases of type I CCs with peculiar morphology. In these cases, the dilatation of the common hepatic and common bile duct was coupled with the dilatation of the central portion of the cystic duct, which rendered the cysts a bicornal configuration. The authors proposed a new subtype of CC, type ID [26]. A few years later, Bhoil et al. suggested a further subdivision of type VI cysts into type VIA, referring to isolated cystic duct cysts, and type VIB, referring to combined dilatation of both the cystic duct and the common bile duct [27]. 

Controversy persists regarding the etiology of CCs. The most widely accepted hypothesis is Babbitt’s theory, which states that CCs are caused by an anomalous pancreaticobiliary ductal junction. This is due to an incomplete migration of the pancreaticobiliary junction, which does not reach the duodenal wall. The common bile duct connects to the pancreatic duct 1–2 cm proximal to the sphincter of Oddi [28]. The resulting common duct measures 10–45 mm in length and is not covered by the sphincter, thus favoring the reflux of pancreatic juice into the common biliary duct. The pancreaticobiliary reflux induces not only increased pressure and subsequent cystic dilation but also, owing to the activated pancreatic enzymes and biliary stasis, choledocholithiasis, formation of protein thrombi, inflammation, epithelial injury, mucosal dysplasia, and eventually, malignancy [5,12,28]. High levels of amylase, trypsinogen, and phospholipase A2 have been detected in CCs, supporting Babbitt’s theory [29,30]. 

Nevertheless, an APBDJ has only been identified in 50–80% of CC cases [31]. Therefore, other etiologic theories have been proposed, such as the obstruction of the distal portion of the common biliary duct, the dysfunction of the sphincter of Oddi, or the paucity of ganglion cells in the distal common biliary duct causing proximal dilation [5].

None of these theories can explain type II CCs, which lack inflammation and portend only slight malignant potential. Furthermore, choledochoceles might arise secondary to the obstruction of the ampulla of Vater [5]. Whether type II and type III CCs are simply biliary duplications cysts and biliary/duodenal duplications cysts, respectively, is still a matter of debate. 

Caroli’s disease, on the other hand, is considered the result of ductal plate malformation [32]. It may also arise in association with autosomal recessive or autosomal dominant polycystic kidney disease (PKHD), caused by the mutation of the PKHD1 gene harbored by chromosome 6p12 [11,32]. 

An APBDJ and its consequences are thought to be the cause of type VI CCs [33,34]. Other factors may contribute, such as an acute angled insertion of the cystic duct into the common bile duct [30] or reduced vascularity that causes wall weakness at the junction between the cystic duct and the common hepatic duct [35]. In addition, recent studies point to an acquired dysfunction of the sphincter of Oddi as the eliciting factor for the development of bile duct cysts [31]. Such dysfunctions are, however, observed in a limited number of cases. 

Hence, the etiology of CCs encompasses a spectrum of congenital and possibly acquired incompletely understood defects.

Typically, patients present the triad of right hypochondriac pain, a palpable abdominal mass, and intermittent jaundice, as well as nausea/vomiting and fever. However, these are not constant findings, and CCs are often asymptomatic in adults, being incidentally diagnosed upon imagistic investigations, which was the case with our patients as well, who only presented with some of the aforementioned symptoms and were initially diagnosed with acute cholecystitis and acute pancreatitis.

Given the unspecific clinical picture, the diagnosis relies on imaging studies. The abdominal ultrasound is highly sensitive for the initial diagnosis of CCs (71–97%) [36], also allowing antenatal diagnosis. It should, however, be completed with an imagistic assessment of the biliary tract and pancreatic duct in order to properly categorize the CC and accurately plan surgical treatment. Technetium-99 hepatobiliary iminodiacetic acid (HIDA) scan has 100% sensitivity for the diagnosis of type I CCs and 67% sensitivity in the case of type IVA CCs and is useful in the evaluation of possible CC rupture [36]. CT is conclusive for the diagnosis of all types of CCs as it reveals the exact extent of both extrahepatic and intrahepatic segment dilations and depicts cyst wall thickening, which raises the suspicion of malignant transformation. Computed tomographic cholangiopancreatography (CTCP) and MRCP are highly effective in outlining the biliary tract, but CTCP is superior for the visualization of the common pancreaticobiliary channel and the bilio-digestive anastomosis after surgery. The most sensitive diagnostic tool remains the ERCP, but the chronic inflammation and scarring hinder the results of the examination [5]. Other drawbacks are its invasive nature, the risk of cholangitis and pancreatitis, and exposure to ionizing radiation. Taking all these into account, MRCP is considered the gold standard for the diagnosis of CCs as it is non-invasive and does not carry the risks associated with ERCP. Usual blood panels may show hepatic cytolysis, cholestasis, leukocytosis in cholangitis, increased serum levels of amylase and lipase in pancreatitis, perturbed coagulation, and abnormal kidney function in severe cases, as well as elevated CA 19-9 values suggestive of malignancy. Our patient had normal CA 19-9 values preoperatively, as well as for all the subsequent follow-up visits.

The histopathological picture of CCs varies according to the patient’s age and cyst type. In children, a scattered columnal/cuboidal epithelium lines the cyst, and its wall displays inflammation and fibrosis. On the other hand, in adults, the histopathological examination of CCs reveals mucosal inflammation and hyperplasia. While types I and IV CCs may present a discontinued or absent biliary mucosal layer, type III CCs may be lined by either duodenal or biliary mucosa [37,38]. In our case, there was remarkable transmural polymorphic inflammatory infiltration with biliary columnar epithelium.

CCs are frequently misdiagnosed. Even with modern imaging techniques, the diagnosis is sometimes challenging, requiring a multidisciplinary effort involving surgeons, radiologists, and gastroenterologists. In many atypical cases, the diagnosis is only made intraoperatively. Differential diagnoses include biliary stricture, choledocholithiasis, hepatic cysts/abscesses, gallbladder/intestinal duplication, duodenal atresia, and mesenteric/omental cysts. 

CCs are associated with a high risk of complications, such as ascending cholangitis; cholelithiasis, cystolithiasis, or hepaticolithiasis; cholecystitis; biliary stricture; recurrent acute pancreatitis; biliary and hepatic cirrhosis; portal vein thrombosis; and malignancy. Such complications are more frequent with age [39].

The reported malignancy rate varies markedly between studies, ranging from 4% to 21.6%, depending on the type of CC. It increases with age, from 0.7% in children aged less than 10 to 14.3% in patients older than 20 [20,40,41]. Among biliary tract cancers, 50–62% are extrahepatic duct cancers, 38–46% are gallbladder cancers, 7% are combined gallbladder cancers and cholangiocarcinomas, and 2.5% are intrahepatic duct cancers [5,42]. Gallbladder carcinoma primarily arises in cases of APBDJ without CCs (up to 50% of cases), given the higher pressures that allow the pancreatic juice to reach the gallbladder and much rarer in APBDJ associated with CCs (5% of cases) [43]. On the other hand, 14% of patients with CCs develop cholangiocarcinoma within the cyst [40]. Adenocarcinomas, adenoacanthomas, squamous cell carcinomas, anaplastic carcinomas, bile duct sarcomas, hepatomas, and pancreatic carcinomas may also develop in association with CC [37]. Although any BC type may trigger malignancy, it most frequently occurs in patients with CCs type I (68% of cases) and IV (21% of cases) [44]. The reported malignancy rates for CCs type II, III, and V are 5%, 1.6%, and 6%, respectively [44]. Only four cases of cholangiocarcinoma complicating type VI CCs have been reported [45]. Malignant transformation may take place within the cyst, the gallbladder, or any other part of the extrahepatic biliary tree [46]. Biliary tract cancers have a very poor prognosis, with the median survival being 6–21 months. Thus, early diagnosis and treatment are essential for a favorable outcome.

Oncogenesis involves several mechanisms, but the main causes are bile stasis and pancreaticobiliary reflux. They generate chronic epithelial inflammation, which, in turn, leads to K-Ras and DPC-4 gene mutations and p53 overexpression, dysplasia/glandular metaplasia, and malignant transformation [47,48]. 

Complete cyst excision and biliary diversion represent the treatment of choice for CCs, either through laparotomy, laparoscopy, or robotic surgery. Minimally invasive surgery carries a substantially lower risk of acute and late complications compared to open surgery [44]. Cyst drainage or incomplete resection should be discarded as the postoperative malignancy risk in these cases exceeds 30% [45]. Surgical reintervention with complete cyst excision is indicated in these cases.

For type I cysts, complete cyst excision is usually followed by Roux-en-Y hepaticojejunostomy (RYHJ), as this is the preferred method of biliary drainage reconstruction. Hepaticoduodenostomy has also been performed, but the risk of complications following this procedure is significantly higher (33%), including biliary and gastric malignancy caused by duodenogastric bile reflux [46,48]. 

Type II biliary cysts may be managed by less aggressive procedures, such as simple diverticulectomy. 

Choledochoceles may only require sphincterotomy, along with a biopsy of the cyst epithelium, in order to rule out dysplasia or cyst marsupialization during ERCP or transduodenally [47]. 

The optimal management of type IVA CCs is controversial. While some experts advocate surgical excision of the extrahepatic biliary tract and hepaticojejunostomy, hepatic resection or liver transplantation are necessary in patients with extensive intrahepatic dilation associated with complications such as cholangitis, lithiasis, or biliary cirrhosis [5,8,20].

While localized Caroli’s disease is successfully managed by segmental hepatectomy, the diffuse disease requires liver transplantation. Palliative measures include percutaneous or endoscopic drainage and stent placement.

In CCs type VIA with a narrow opening into the common bile duct that shows no pathological histological features, cholecystectomy and excision of the cystic duct usually suffice. Cases of type VIA cysts with a wide opening into the common bile duct and those with a histological picture typical for a choledochal cyst, as well as type VIB cysts, require cholecystectomy, complete excision of the common bile duct and Roux-en-Y hepatico-jejunostomy [24,49]. 

Even in the absence of CCs, cholecystectomy and complete excision of the common bile duct are recommended in patients with APBDJ, given the significant risk of gallbladder cancer discussed above [22]. 

Unfortunately, despite radical treatment, early postoperative complications can occur and include ileus, bile leakage, pancreatitis, and pancreatic fistula. Late complications are rare, but tedious and multiple issues have been reported, from incisional hernias to biliary stricture, choledocholithiasis, and recurrent cholangitis, while between 0.7 and 10% of patients develop cholangiocarcinoma or pancreatic adenocarcinoma after surgery at variable intervals, even after several decades [10,13,50,51,52]. The largest retrospective study published so far reports on the long-term follow-up of 3911 patients who underwent CC resections, and the authors propose a life-long evaluation of patients, with an increase in the frequency of liver function tests and ultrasound assessments after the first 20 years following the initial resection [53]. ERCP carries associated risks that have been exhaustively reported in the literature. However, in cases where there is doubt regarding the type of CC when the patient’s status permits preoperative preparation (such as the implantation of a stent to reduce bilirubin levels) and the multidisciplinary team deems this necessary, and postprocedural surgery is planned, the benefits of performing diagnostic and/or therapeutic ERCP should be weighed against the risks. Surgical treatment performed at a later age is associated with an increased frequency of such complications due to the chronic inflammatory and fibrotic changes in the cyst wall; therefore, a team comprising surgeons, gastroenterologists, radiologists, and intensive care specialists should approach each case individually and implement a tailored approach. Incomplete excision remains the main risk factor for malignant transformation; therefore, the importance of accurate imagistic diagnosis and careful preoperative planning cannot be emphasized enough. 

## 4. Conclusions

CCs are rare, premalignant anomalies of the biliary tract that present non-specific signs and symptoms. The diagnosis requires a high index of suspicion and thorough imaging studies. Early diagnosis and treatment are crucial for the avoidance of complications, the most severe of which is a malignant transformation that may involve the biliary ducts, the gallbladder, or the pancreas. Precise preoperative diagnosis is often very challenging, even with modern imaging techniques. Long-term postoperative monitoring is mandatory as life-threatening complications, including malignancy, may occur even decades after the intervention.

Awareness of both surgeons and radiologists of the numerous variants of CCs that are not included in the generally used Todani’s classification is of great importance for an accurate preoperative diagnosis and, implicitly, correct management. Therefore, we find the revision of this classification urgently necessary.

## Figures and Tables

**Figure 1 diagnostics-13-01059-f001:**
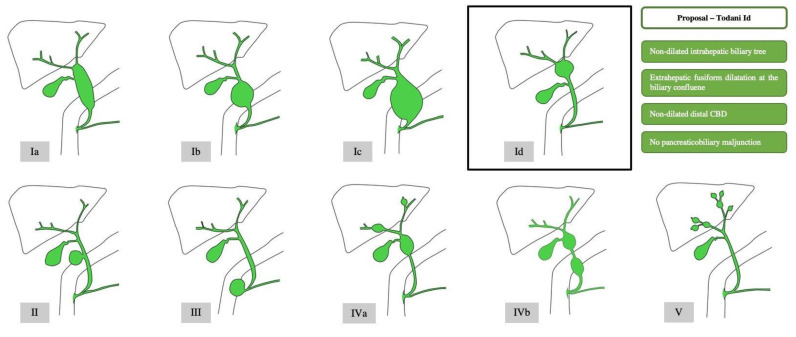
Todani classification of choledochal cysts, with proposed subtype ID.

**Figure 2 diagnostics-13-01059-f002:**
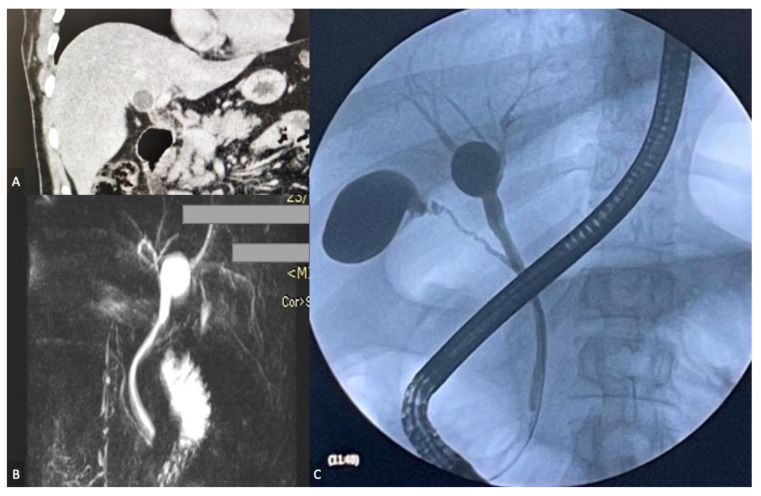
(**A**) CT scan, coronal section—type II CC; (**B**) MRCP reconstruction—type II CC; (**C**) ERCP—fusiform dilatation of the biliary confluence.

**Figure 3 diagnostics-13-01059-f003:**
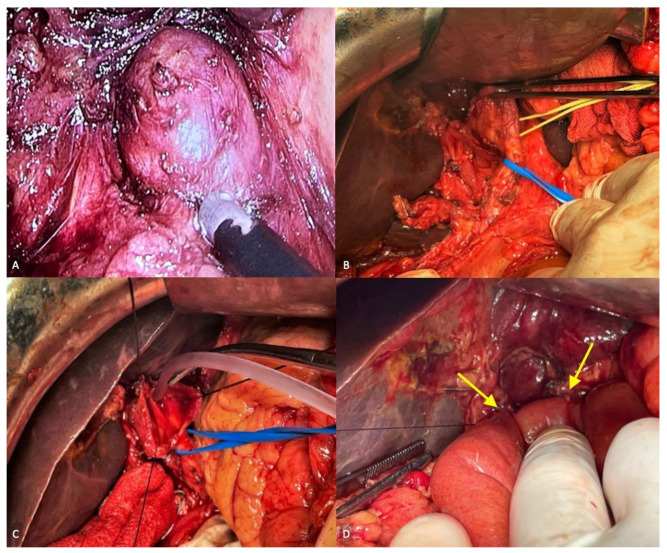
(**A**) laparoscopic dissection of the hilar CC; (**B**) circumferential dissection of the CC; (**C**) opening of the CC and cannulation of the right and left hepatic ducts; (**D**) complete double-tutorized hepaticojejunostomy.

**Figure 4 diagnostics-13-01059-f004:**
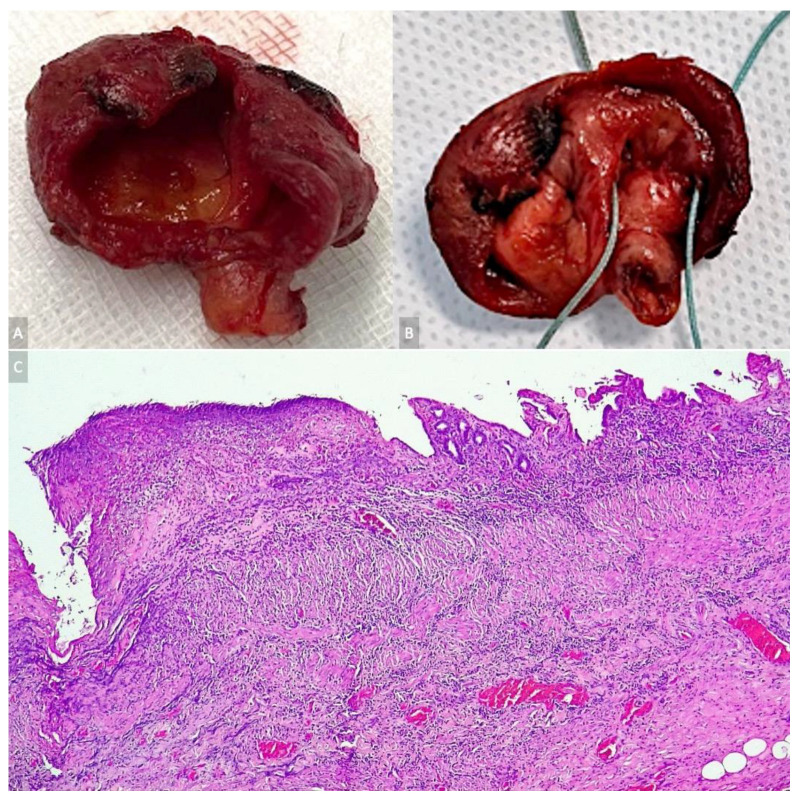
(**A**,**B**) Thick-walled fusiform dilatation of the biliary confluence; (**C**) biliary columnar epithelium of the CC with focal ulcerations and marked transmural polymorphic inflammatory infiltration.

## Data Availability

The data presented in this study are available on request from the corresponding author. The data are not publicly available due to privacy restrictions.

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
