# Peer review of "The Curious Case of the Choledochal Cyst—Revisiting the Todani Classification: Case Report and Review of the Literature"

_diagnostics, 2023, doi:10.3390/diagnostics13061059_

Round 1

Reviewer 1 Report

This review represents a commendable work by Miron and collegueas on a neglected topic. In fact, papers about choledocal cysts are not so frequent due the rarity of the pathology, so I think that a review would be welcome. 

I have some comments.

- It would be very useful if the authors could produce some outlines (figures) of the cysts classification, even hand drawn or cited by other authors. The new, proposed variant of cyst discussed in the case report by the authors could be more clear and highlighted. 

- Please explain in the case report why an ERCP was performed. If there was a specific diagnostic need or a therapeutic proposal please explain which it was. Otherwise, I would enhance in the case report and in the discussion that ERCP must be only therapeutic due its potential complications. 

- In the discussion, I would add a comment about potential biliary complications after surgery (i.e. cite Ray et al Am J Surgery 2023) and if there are some cases or series of recurrence. 

Author Response

Hello, 

Thank you very much for your review and for your appreciation and valuable suggestions. 

We have made the following additions/changes to the manuscript:

  • We added a figure outlining the existing types of choledochal cysts, as well as the variant we encountered and proposed as Id. We also underlined this aspect as suggested. 

  • We explained why we deemed ERCP necessary - imaging studies suggested a type II CC and we had planned to implant a biliary stent that would help in the subsequent laparoscopic resection; the ERCP proved extremely useful, as it was the only preoperative study that portrayed the actual aspect of the CC and changed our approach entirely. While it is true that diagnostic ERCP carries the risk of complications, we strongly believe that in such cases, in which post-procedural surgery is planned either way, it is useful and should be performed after a multidisciplinary team evaluates each case. CC have a relatively high risk of malignancy, especially types I and IV (as shown in the text), and accurate diagnosis helps tremendously in short and long-term case management. 

  • We added the suggested reference, as well as two other case series and a systematic review that report long-term complications following surgery for CC. We found no cases describing cystic recurrence, but other complications were reported with different rates of frequency (leakage, pancreatitis, recurrent cholagitis, stricture, malignancy).

Thank you for taking the time to review our article and we hope the changes we made are satisfactory.

Reviewer 2 Report

The authos have described the CC located at the hepatic hilar. However, they did not mention about pancreaticobiliary maljunction, I think there is no PM in this case in MRCP. From these background, this CC seemed to be type V located at the hepatic hilar. 

Author Response

Hello, 

Thank you for your review and for your suggestions. 

We have made the following additions/changes to the manuscript:

  • We added a figure outlining the existing types of choledochal cysts, as well as the variant we encountered and proposed as Id. As shown in the figures, as well as the imaging studies and intraoperative findings, there was only one choledochal cyst, with the rest of the biliary tree entirely normal, including the intrahepatic biliary ducts, cystic duct, common bile duct below the cyst, and the pancreaticobiliary junction, which was indeed normal. It is definitely not a type V cyst, which is Caroli disease - multifocal cystic transformation of the intrahepatic biliary tree.  
  • We supplemented the information presented in the introduction, results and discussion sections. 

We hope you find these changes satisfactory.

Thank your for taking the time to review our article!

Round 2

Reviewer 2 Report

the authors sincerely answered to the reviewer's suggestions.